# Agentic RL Scaling Law: Spontaneous Code Execution for Mathematical Problem Solving

**Xinji Mai**[1,2,†]    **Haotian Xu**[2,†]    **Xing W**[2]    **Weinong Wang**[2]
**Yingying Zhang**[3,*]    **Wenqiang Zhang**[1,4,*]

[1]College of Intelligent Robotics and Advanced Manufacturing, Fudan University
[2]Xiaohongshu
[3]East China Normal University
[4]Shanghai Key Lab of Intelligent Information Processing,
College of Computer Science and Artificial Intelligence, Fudan University
`xjmai23@m.fudan.edu.cn,{xuhaotian,wuxing,wangweinong}@xiaohongshu.com`

## Abstract

Large Language Models (LLMs) often struggle with mathematical reasoning tasks requiring precise, verifiable computation. While Reinforcement Learning (RL) from outcome-based rewards enhances text-based reasoning, understanding how agents autonomously learn to leverage external tools like code execution remains crucial. We investigate RL from outcome-based rewards for Tool-Integrated Reasoning, ZeroTIR, training base LLMs to spontaneously generate and execute Python code for mathematical problems without supervised tool-use examples. Our central contribution is we demonstrate that as RL training progresses, key metrics scale predictably. Specifically, we observe strong positive correlations where increased training steps lead to increases in the spontaneous code execution frequency, the average response length, and, critically, the final task accuracy. This suggests a quantifiable relationship between computational effort invested in training and the emergence of effective, tool-augmented reasoning strategies. We implement a robust framework featuring a decoupled code execution environment and validate our findings across standard RL algorithms and frameworks. Experiments show ZeroTIR significantly surpasses non-tool ZeroRL baselines on challenging math benchmarks. Our findings provide a foundational understanding of how autonomous tool use is acquired and scales within Agent RL, offering a reproducible benchmark for future studies. Code is released at https://github.com/yyht/openrlhf_async_pipline.

## 1  Introduction

LLMs have demonstrated remarkable capabilities across various domains. However, they often face challenges when tackling tasks that demand precise, multi-step reasoning and complex computations, particularly within the realm of mathematics[1, 2]. The inherent nature of next-token prediction often leads LLMs to generate responses based on high likelihood rather than computational correctness, hindering their reliability for mathematical problem-solving. Existing approaches to augment the mathematical abilities of LLMs typically involve Supervised Fine-Tuning (SFT) on specific datasets or integrating external tools in a controlled manner[3]. SFT, while potentially effective, often necessitates extensive, high-quality trajectory data and may constrain the model's capacity to explore novel problem-solving strategies, potentially leading to overfitting on specific solution patterns and sacrificing generalizability. Tool-Integrated Reasoning (TIR) methods, such as the TIR capability mentioned in the context of Qwen2.5-Math [4], usually depend on specific prompt structures or

---

† Equal contribution.    * Corresponding authors.

39th Conference on Neural Information Processing Systems (NeurIPS 2025).

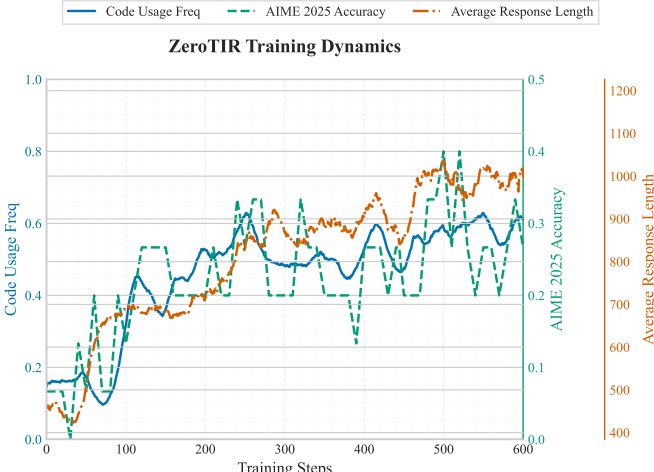

Figure 1: Illustration of Agent RL Scaling Law during Zero-TIR training. The observed trends, including the non-monotonic scaling of code usage and the positive correlation between training steps, code usage, response length, and task accuracy, exemplify the core scaling dynamics.

predefined triggers to invoke tools like code interpreters. This paradigm differs fundamentally from fostering an agent that learns to utilize tools *spontaneously* based on the emergent needs of the task.

The advent of models like DeepSeek-R1 [5] has highlighted the significant success of RL in scaling LLM reasoning capabilities directly from base models, which is also known as ZeroRL. This approach enables emergent abilities, such as self-correction and reflection, using only outcome-based rewards. Works like Open DeepResearch [6] have underscored the immense potential of tool invocation within LLMs. However, the application of agentic tool use, particularly code execution for mathematical tasks, has received comparatively less attention. We posit that for mathematical problems, precise and deterministic computation via code execution is often more advantageous than retrieving potentially noisy textual information from search engines, primarily due to the deterministic nature and rapid feedback loop of code execution environments[7, 8]. We also note contemporaneous work, such as the excellent contribution by TORL [9], which explores the potential of using code execution for mathematical tasks.

However, we believe that RL from a fine-tuned model with existing tooling capabilities obscures some important findings. Similar to RL from a model after SFT, it is difficult to observe a relationship between response length and performance. The paper aims to provide a more comprehensive and clearer analysis to facilitate community research and reproduction of what we term Agent RL Scaling Law. We present exhaustive experiments conducted using mainstream community frameworks (Open-Reasoner-Zero [10], OpenRLHF [11]) and popular RL algorithms (PPO [12], Reinforce++ [13]), coupled with an environment server. We investigate how LLMs, initialized from base models, can spontaneously learn to leverage a Python code execution environment through RL. Our central hypothesis is that the learning process for utilizing such a tool adheres to discernible patterns, which we designate as Agent RL Scaling Law.

- We identify and characterize novel Agent RL Scaling Law that govern the autonomous acquisition of spontaneous code execution skills in ZeroTIR for mathematical reasoning.
- We propose and implement an effective framework, ARL, for training base LLMs to spontaneously leverage code execution, which can be quickly enabled on the community mainstream RL training frameworks.
- We provide extensive empirical validation showing that ZTRL model trained with ZeroTIR significantly outperform non-instrumental ZeroRL baselines on challenging mathematical benchmarks and SFT-based TIR methods.

By discovering the Agent RL Scaling Law and developing a robust framework to leverage them, our approach significantly advances the understanding and application of reinforcement learning for training capable LLM agents that autonomously utilize tools for complex reasoning tasks like mathematics.

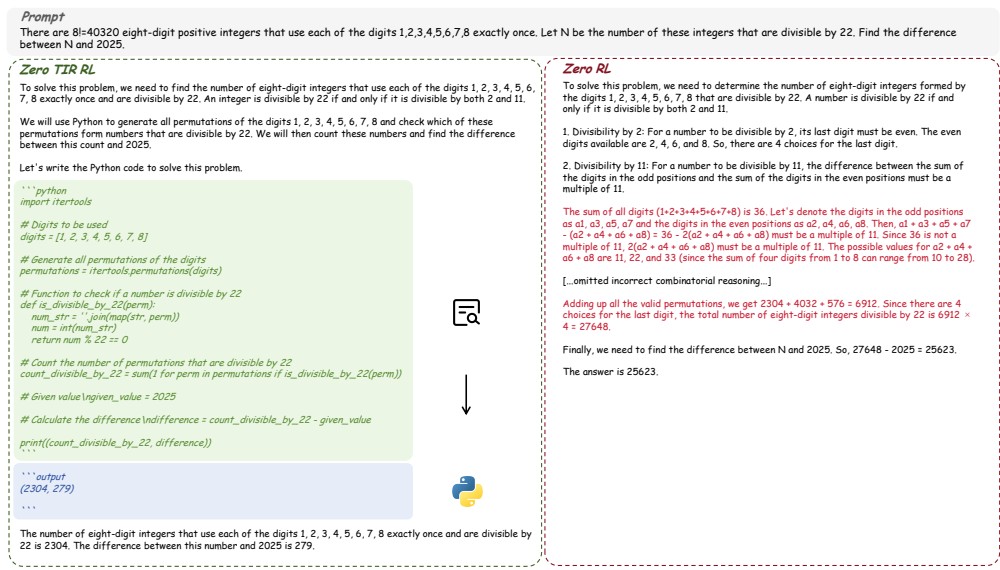

Figure 2: Example responses to the same prompt, contrasting the code-augmented solution generated by ZeroTIR (left) with the reasoning-only solution from ZeroRL (right).

## 2 Related Works

Our research builds upon advancements in LLM reasoning, TIR, and agent reinforcement learning.

### 2.1 Tool-Integrated Reasoning

To overcome the limitations of purely internal reasoning, particularly in domains requiring calculation or factual look-up[14, 15], TIR approaches augment LLMs with external tools. These tools can range from calculators and search engines to code interpreters. Much existing work implements TIR through SFT, training models on datasets containing tool invocation examples, or through carefully designed prompts and controlled mechanisms that trigger tool use under specific conditions, such as Qwen2.5-Math-TIR [4, 16]. These methods typically guide or compel the model to use tools according to predefined patterns or explicit instructions[17]. Our research diverges from these controlled approaches by focusing on spontaneous tool use. We investigate whether an agent can learn autonomously, through ZeroRL, when and how to utilize a code execution tool based purely on maximizing task success reward, without being explicitly instructed or pre-trained on tool-use trajectories.

### 2.2 Agent Reinforcement Learning

LLM Agents, capable of autonomous planning, decision-making, and environmental interaction including tool use, represent a significant evolution [18]. Reinforcement Learning provides a powerful paradigm for training these agents. Agent RL has been applied to train LLMs for information retrieval tasks, with frameworks like Search-R1 [19] and R1-Searcher [20] teaching models to autonomously query search engines during reasoning using outcome-based rewards[21]. This principle is also being applied to computational tools; Contemporaneous work like TORL [9] uses ZeroRL to train agents for code interpreter use in mathematics. A notable trend in Agent RL is the effectiveness of simpler, outcome-based rewards over complex process rewards or imitation learning, fostering exploration and the emergence of novel strategies. We situate our research within the Agent RL landscape, specifically focusing on the integration of a code execution tool for enhancing mathematical reasoning of base model. Distinct from search-focused agents and building upon the ZeroRL paradigm, our primary contribution lies in the systematic identification and analysis of the Agent RL Scaling Law that govern the learning dynamics of spontaneous tool acquisition across various RL algorithms and frameworks, providing a foundational understanding and reproducible benchmark for this specific type of Agent RL.

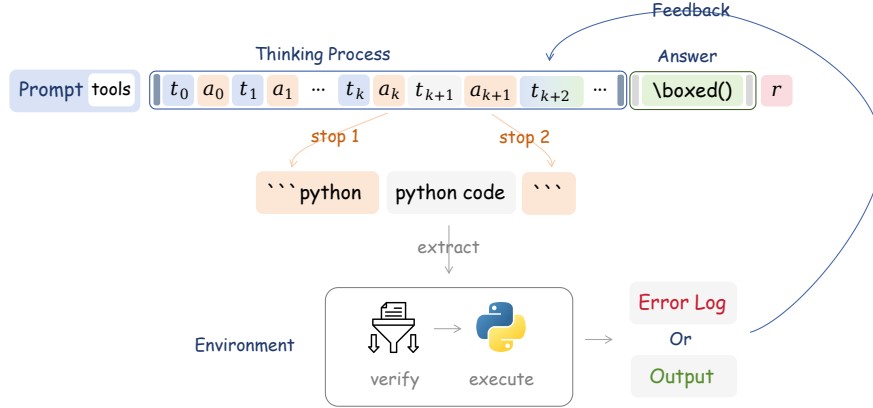

Figure 3: Detailed schematic of the interactive rollout process.

# 3 Methodology

ZeroTIR, trains a base LLM to autonomously utilize a Python code execution environment for mathematical problem-solving through reinforcement learning. This section details the core components of our framework, including the RL formulation, the interaction mechanism between the LLM and the code environment, and specific design choices made during implementation and training.

## 3.1 RL for Spontaneous Code Execution

Our methodology utilizes reinforcement learning to train the LLM agent, primarily employing policy gradient algorithms such as PPO and REINFORCE variants like Reinforce++. PPO operates as an actor-critic method, optimizing a policy network $\pi_\theta$ while concurrently training a value network $V_\phi$ to estimate state values, which aids in reducing the variance of policy gradient estimates. The policy update in PPO typically maximizes a clipped surrogate objective function:

$$L^{CLIP}(\theta) = \mathbb{E}_t \left[ \min \left( r_t(\theta)\hat{A}_t, \text{clip}(r_t(\theta), 1 - \epsilon, 1 + \epsilon)\hat{A}_t \right) \right]$$

where $r_t(\theta) = \frac{\pi_\theta(a_t|s_t)}{\pi_{\theta_{old}}(a_t|s_t)}$ is the probability ratio between the current policy and the policy used to collect the data ($\pi_{\theta_{old}}$), $\hat{A}_t$ is the estimated advantage at timestep $t$, and $\epsilon$ is a hyperparameter defining the clipping range. The value function $V_\phi$ is trained concurrently, usually by minimizing the mean squared error against target values derived from sampled rewards.

A crucial aspect of applying PPO in our setting involves handling the tokens inserted by the external code execution environment $\mathcal{E}_{code}$. The value function $V_\phi(s_t)$ should not be trained on states $s_t$ corresponding to these environment-generated tokens, as they are not products of the policy $\pi_\theta$. Thus, the value loss computation must mask these states. Furthermore, the advantage estimation $\hat{A}_t$, especially when using Generalized Advantage Estimation (GAE) with $\lambda \neq 1$, requires careful consideration. GAE computes advantages based on temporal difference (TD) errors:

$$\hat{A}_t^{GAE} = \sum_{l=0}^{T-t-1} (\gamma\lambda)^l \delta_{t+l}, \quad \text{where} \quad \delta_t = r_t + \gamma V_\phi(s_{t+1}) - V_\phi(s_t)$$

Here, $\gamma$ is the discount factor and $\lambda$ is the GAE parameter balancing bias and variance. Since $V_\phi(s_{t+1})$ is unreliable or ill-defined if $s_{t+1}$ consists of environment feedback tokens, terms involving these states can introduce noise or bias into $\hat{A}_t^{GAE}$. A common practice is to effectively mask the influence of these tokens, potentially by setting $\hat{A}_t = 0$ at positions corresponding to environment-inserted tokens during the policy loss calculation, preventing spurious updates to $\pi_\theta$. However, we found that setting $\lambda$ to 1 did not affect the results, so here we set $\lambda$ to 1.

REINFORCE-style algorithms, including Reinforce++, operate differently, typically foregoing an explicit learned value function. The policy gradient is estimated directly using sampled trajectories.

The general form of the policy gradient objective is:

$$\nabla_\theta J(\theta) = \mathbb{E}_{\tau \sim \pi_\theta} \left[ \sum_{t=0}^{T-1} \nabla_\theta \log \pi_\theta(a_t|s_t) \hat{A}_t \right]$$

where the agent learns by adjusting $\theta$ in the direction suggested by the gradient. The advantage estimate $\hat{A}_t$ in this context is often calculated using the discounted Monte Carlo return $G_t$ minus a baseline $b(s_t)$:

$$\hat{A}_t = G_t - b(s_t), \quad \text{where} \quad G_t = \sum_{k=t}^{T} \gamma^{k-t} r_k$$

The baseline $b(s_t)$ serves to reduce variance and can be a simple running average of returns or potentially more sophisticated estimates, but crucially, it doesn't rely on a learned value function $V_\phi$ in the same way PPO does. While the advantage calculation itself is less directly complicated by the value of environment states, careful reward attribution and ensuring only policy-generated actions ($a_t$ corresponding to LLM tokens) contribute to the gradient term $\nabla_\theta \log \pi_\theta(a_t|s_t)$ remain essential. Reinforce++ specifically may incorporate advanced baseline techniques or modifications to the update rule for improved performance compared to vanilla REINFORCE.

Irrespective of the specific algorithm, the overall learning objective remains the maximization of the expected outcome-based reward $R(x, y)$, regularized by the KL divergence from a reference policy $\pi_{ref}$ to maintain stability:

$$\max_\theta \mathbb{E}_{x \sim \mathcal{D}, y \sim \pi_\theta(\cdot|x; \mathcal{E}_{code})}[R(x, y)] - \beta D_{KL}[\pi_\theta(\cdot|x; \mathcal{E}_{code})||\pi_{ref}(\cdot|x; \mathcal{E}_{code})]$$

Here, $x$ is the input problem, $y$ is the full trajectory including interactions with the code environment $\mathcal{E}_{code}$, $R(x, y)$ is the outcome-based reward, $\pi_{ref}$ is the reference policy, $D_{KL}$ is the KL divergence, and $\beta$ controls regularization.

### 3.2 Training Stability and Efficiency Techniques

Training stability and efficiency are crucial in interactive RL settings like ZeroTIR. To address potential instability, such as training collapse observed in some ZeroRL frameworks, we employ two key techniques.

First, we introduce a replay buffer filtering mechanism to enhance stability and focus learning. Multiple responses generated for the same prompt are grouped, and their final answer accuracy (based on outcome rewards) is calculated. We filter out groups with accuracy above a high threshold 0.8 or below a low threshold 0.2, prioritizing samples within the intermediate range where the learning gradient is likely most beneficial.

Second, we implement an efficient interaction mechanism for spontaneous code execution during rollouts, depicted schematically in Figure 3 and detailed in Algorithm 1. This method leverages dynamic stop tokens (e.g., "'python, "') to iteratively manage reasoning, code generation, interaction with the external code environment, and integration of execution feedback. This state-machine approach is significantly more efficient than generating complete sequences followed by post-hoc parsing for code extraction. This mechanism also facilitates managing tool interaction frequency by counting completed execution cycles ($n_{calls}$). For experimental control, particularly in initial runs managing computational resources, we enforce a maximum call limit ($N_{max}$). Upon reaching this limit, a notification ("Tool call count has been exhausted. You can no longer call the tool.") is injected into the context before the final generation resumption, ensuring the agent relies on internal reasoning thereafter.

### 3.3 Environment Interaction Frameworks

Effective Agent RL involving external tools necessitates robust and scalable environment interaction. Integrating execution environments directly into RL training code often creates dependency issues and hinders modularity. Our initial approach addressed this by implementing the Python code execution environment as an independent, network-accessible service. This decoupled architecture enhances robustness compared to embedding the interpreter locally, as service failures do not crash the main training process. It also improves maintainability and allows the service to be scaled

**Algorithm 1** ZeroTIR Rollout with Spontaneous Code Calls

**Require:** policy $\pi$, prompt $P$, code env $E$, call budget $N$
1:  $C \leftarrow P,\ T \leftarrow \emptyset,\ k \leftarrow 0$
2: **while true do**
3:     $(s, \sigma) \leftarrow \pi.\text{GENERATE}(C, \{\text{EOS}, /boxed\{\}, \text{```python```}\})$
4:     $C, T \leftarrow C + s,\ T + s$
5:     **if** $\sigma \in \{\text{EOS}, /boxed\{\}\}$ **then return** $T$
6:     **end if**
7:     **if** $\sigma = \text{```python}$ **then**
8:       $k \leftarrow k + 1$
9:       **if** $k > N$ **then return** $T$
10:      **end if**
11:      $(c, \_) \leftarrow \pi.\text{GENERATE}(C, \{\text{```}\})$
12:      $C, T \leftarrow C + c,\ T + c$
13:      $r \leftarrow E.\text{EXEC}(\text{EXTRACT}(c))$
14:      $C, T \leftarrow C + \text{FMT}(r),\ T + \text{TOKENS}(r)$
15:     **elsereturn** $T$
16:     **end if**
17: **end while**

Table 1: Performance comparison on key mathematical reasoning benchmarks.

| Model | Params | Tool | AIME24 | AIME25 | MATH500 | Avg. | Code Prop. |
|---|---|---|---|---|---|---|---|
| Qwen2.5 Ins. | 7B | ✗ | 13.3% | 20.0% | 75.8% | 36.4% | 0.0 |
| Qwen2.5 Ins. | 7B | ✓ | 16.7% | 0.0% | 76.4% | 31.0% | 0.0 |
| Qwen2.5 Math Ins. | 7B | ✗ | 13.3% | 6.7% | 83.2% | 34.4% | 0.0 |
| Qwen2.5 Math Ins. | 7B | ✓ | 20.0% | 26.7% | 78.0% | 41.6% | **95%** |
| SimpleRL-Zero | 7B | ✗ | 33.3% | 6.7% | 77.2% | 39.1% | 0.0 |
| rStar-Math | 7B | ✗ | 26.7% | - | 78.4% | 52.6% | 0.0 |
| Eurus-2-PRIME | 7B | ✗ | 26.7% | 13.3% | 79.2% | 39.7% | 0.0 |
| TORL | 7B | ✓ | 43.3% | 30.0% | 82.2% | 51.8% | 83% |
| ZTRL | 7B | ✓ | **46.7%** | **30.0%** | **85.2%** | **54.0%** | 90% |

independently using standard web serving technologies like Flask, Gunicorn for concurrency, Nginx for load balancing, and client-side rate limiting with libraries such as 'aiolimiter' to handle numerous concurrent requests stably. The RL framework interacts with this service via standard HTTP calls.

While the decoupled service provides stability and scalability, synchronous interaction can become a bottleneck. To further improve training throughput, especially for large-scale experiments, we implemented an enhanced interaction framework within OpenRLHF based on asynchronous rollout and pipelining. In this setup, each rollout actor manages an asynchronous queue, which is pre-filled with generated experiences. During training, data retrieval from a queue occurs concurrently with the submission of a new rollout task to an actor. Experience generation proceeds asynchronously while the trainer performs parameter updates, and results are collected afterward to refill the queue. An adaptive mechanism handles potential data shortages in the replay buffer due to filtering by synchronously collecting completed rollouts when needed, preventing excessive data accumulation at the rollout actors. This asynchronous pipeline approach yielded significant speedups in our experiments, proving approximately 1.6 times faster than basic asynchronous rollout and over 4 times faster than the initial synchronous decoupled server interaction, enabling more extensive experimentation, particularly with larger models. For detailed design, please refer to our github repository.

## 4 Experiments

In this section, we detail the experimental setup designed to evaluate our ZeroTIR approach and validate the existence of Agent RL Scaling Law for spontaneous code execution in mathematical reasoning. We describe the datasets, baselines, implementation details, and present the core results demonstrating the effectiveness of our method and the observed scaling phenomena.

## 4.1 Experimental Setup

Our experiments primarily utilize the Qwen 2.5 Base 7B/32B model, starting directly from pre-trained weights to align with the ZeroRL philosophy. We implement our ZeroTIR approach using standard community frameworks OpenRLHF and Open-Reasoner-Zero, and evaluate key RL algorithms including PPO and Reinforce++. The training dataset consists of ORZ-57k[22] and deepmath[23] dataset containing verifiable mathematical problems. We call the model trained in this way ZTRL. Model performance is evaluated on a suite of standard mathematical reasoning benchmarks such as MATH500[24], AIME24/25[25, 26], HMMT Feb. 24/25[27], cmimc[28], olymemath[29] and so on, which are some of the most difficult mathematical data sets out there.

Key RL hyperparameters include a rollout batch size of 128, with 16 samples generated per prompt. We use 1 policy update step and 12 critic update steps per iteration. Micro-batch sizes for training and forward passes are set to 1. Stability and efficiency techniques, including group-accuracy replay buffer filtering and dynamic stop-token based interaction (detailed in Section 3.2), are employed. The decoupled code execution environment (Section 3.3) handles all tool calls. For initial scaling law validation experiments, the maximum tool calls per trajectory were limited ($N_{max} = 20$) for efficiency.The evaluation metrics include greedy decoding (temperature=0), majority voting, pass@k, and the final performance measured under different top-p sampling settings (temperature=1).

## 4.2 Comparative Performance Analysis

Only Qwen base, beyond Qwen math. To evaluate the effectiveness of our ZeroTIR approach, denoted ZTRL, we present a comparative analysis in Table 1. This table compares our method, trained on the Qwen 2.5 Base 7B model, against relevant baselines and state-of-the-art models[30–32].

The results clearly demonstrate the significant advantage of our ZeroTIR approach. Our 7B ZTRL model achieves a strong average performance of 52.3% across AIME24, AIME25, and MATH500. Furthermore, ZTRL at 52.3% average significantly surpasses the performance of the Instruct model using prompted TIR, and other ZeroRL methods without tool integration like SimpleRL-Zero at 39.1% and Eurus-2-PRIME at 39.7%. Notably, our ZTRL model, trained from the general Qwen 2.5 Base, also outperforms the specialized Qwen 2.5 Math Instruct model when used with its integrated TIR capability, achieving a 52.3% average compared to 41.6%. We also compare our results with TORL [9], significant concurrent work applying ZeroRL with TIR to mathematical reasoning. It is important to note that the reported TORL results utilized the math-specialized Qwen 2.5 Math Base model for training. As shown in Table 1, our ZTRL approach, despite starting from the general Qwen 2.5 Base model, achieves a slightly higher average score of 52.3% versus TORL's 51.8% on the AIME24, AIME25, and MATH500 benchmarks. This highlights the robustness and effectiveness of our specific ZeroTIR implementation in eliciting strong tool-augmented reasoning capabilities even without a domain-specialized base. The high code usage proportion observed for our model, 89%, is comparable to TORL's 83% and correlates strongly with the achieved performance, consistent with the Agent RL Scaling Law discussed.

## 4.3 Analysis of hyperparameters

Table 2 confirms a strong monotonic relation between the interaction cap $N_{max}$ and accuracy across all model scales. Raising the cap from zero to four or twenty lifts average scores by as much as fifteen percentage points, though gains taper beyond four calls, underscoring an agent reinforcement-learning scaling law in which additional tool use yields better problem solving. Performance also grows with model size. Under identical hyperparameters the 32B model surpasses the 7B variant, which itself exceeds the 1.5B baseline. The code-usage ratio rises non-linearly, implying that larger models either resolve more tasks without code or employ code more efficiently. At the 7B scale Reinforce++ and PPO reach similar final accuracies, yet Reinforce++ converges roughly three hundred steps sooner, attaining near-optimal performance around step four hundred while PPO needs more than seven hundred. DeepMath training offers a small but consistent edge over Orz-57k, so Reinforce++ with DeepMath was selected for 32B runs to combine efficiency and ceiling.

Table 4.3 shows that data choice still matters at high capacity. DeepMath delivers the highest Max score, sixty percent on HMMT Feb. 25, whereas Orz-trained models provide stronger majority robustness on CMIMC, fifty-three percent Maj versus thirty-three for DeepMath. Proof-focused curricula sharpen peak reasoning, while heterogeneous contest data stabilises consensus. Decoding

Table 2: Detailed final performance comparison on selected mathematical reasoning benchmarks.

| Method | Params | Algorithm | Dataset | Train Env. Iter. | Eval Env. Iter. | aime25 | aime24 | hmmt feb. 25 | hmmt feb. 24 | cmimc | olymemath | math500 | avg. | code ratio |
|---|---|---|---|---|---|---|---|---|---|---|---|---|---|---|
| ZTRL | 1.5B | ppo | orz-57k | 0 | 0 | 10.0% | 3.3% | 0.0% | 0.0% | 3.3% | 2.0% | 55.8% | 10.6% | 0.000 |
| ZTRL | 1.5B | ppo | orz-57k | 2 | 2 | 3.3% | 3.3% | 0.0% | 0.0% | 3.3% | 1.25% | 60.6% | 10.3% | 0.073 |
| ZTRL | 1.5B | ppo | orz-57k | 4 | 4 | 10.0% | 20.0% | 10.0% | 0.0% | 10.0% | 5.0% | 59.4% | 16.3% | 2.161 |
| ZTRL | 1.5B | ppo | orz-57k | 20 | 20 | 13.3% | 13.3% | 10.0% | 0.0% | 13.3% | 7.75% | 62.6% | 17.2% | **4.090** |
| ZTRL | 7B | ppo | orz-57k | 0 | 4 | 26.7% | 13.3% | 13.3% | 6.7% | 10.0% | 8.2% | 80.6% | 22.7% | 0.143 |
| ZTRL | 7B | ppo | orz-57k | 20 | 20 | 26.7% | 50.0% | 10.0% | 20.0% | 16.7% | 13.5% | 80.2% | 31.0% | 3.490 |
| ZTRL | 7B | Reinforce++ | orz-57k | 2 | 2 | 26.7% | 30.0% | 16.7% | 13.3% | 26.7% | 12.3% | 82.8% | 29.7% | 3.686 |
| ZTRL | 7B | Reinforce++ | deepmath | 2 | 2 | 16.7% | 36.7% | 20.0% | 10.0% | 20.0% | 13.5% | 81.0% | 29.6% | 1.710 |
| ZTRL | 7B | Reinforce++ | deepmath | 2 | 4 | 16.7% | 40.0% | 16.7% | 16.7% | 20.0% | 13.2% | 80.6% | 29.1% | 2.417 |
| ZTRL | 7B | Reinforce++ | deepmath | 4 | 2 | 26.7% | 36.7% | 16.7% | 23.3% | 20.0% | 12.7% | 81.2% | 31.3% | 2.257 |
| ZTRL | 7B | Reinforce++ | deepmath | 4 | 4 | 26.7% | 33.3% | 20.0% | 23.3% | 20.0% | 12.5% | 82.0% | 32.1% | 2.470 |
| ORZ | 7B | PPO | orz-57k | 0 | 0 | 10.0% | 16.7% | 0.0% | 6.7% | 10.0% | 7.0% | 82.2% | 18.9% | 0.000 |
| DeepMath-Zero | 7B | / | deepmath | 0 | 0 | 13.3% | 23.3% | 13.3% | 6.7% | 10.0% | 7.2% | 82.4% | 22.3% | 0.000 |
| ORZ | 32B | PPO | orz-57k | 0 | 0 | 30.0% | 40.0% | 20.0% | 20.0% | 30.0% | 20.8% | 90.6% | 35.9% | 0.000 |
| ZTRL | 32B | Reinforce++ | deepmath | 2 | 2 | 26.7% | 53.3% | 20.0% | 16.7% | 20.0% | 16.7% | 86.2% | 34.2% | 1.691 |
| ZTRL | 32B | Reinforce++ | deepmath | 2 | 4 | 26.7% | 50.0% | 16.7% | 26.7% | 23.3% | 19.0% | 87.8% | 35.7% | 1.994 |
| ZTRL | 32B | Reinforce++ | deepmath | 4 | 2 | **33.3%** | **56.7%** | **20%** | **26.7%** | 33.3% | 17.5% | 87.8% | **39.3%** | 1.558 |
| ZTRL | 32B | Reinforce++ | deepmath | 4 | 4 | 30.0% | 46.7% | 20.0% | 23.3% | **36.7%** | **21.8%** | **89.4%** | 38.2% | 1.863 |

entropy drives a clear Max–Maj trade-off. With a four-call budget and DeepMath training, raising top-$p$ from zero point seven to one point zero increases Max by seven points on HMMT Feb. 25 yet lowers Maj and Avg on several sets. Reduced entropy boosts agreement with only modest loss of peak score. Lowering the training interaction cap from four to two seldom harms and sometimes improves evaluation accuracy at cap four. On AIME25 Max rises from fifty to sixty-six percent, indicating that the large model quickly internalises useful code heuristics and that excessive training calls may overfit early exploration.

These findings show that very large models benefit less from extreme training interaction budgets, that dataset diversity and decoding entropy must be tuned jointly to balance peak and robust metrics, and that curriculum choice remains influential even at the 32B scale.

Table 3: Four training configurations (step = 350) across five math-contest datasets. Train/Test = interaction limit; orz = orz-57k; dm = deepmath; Max = max@32; Maj = maj@32; P@1 = pass@1; Avg = avg@32.

| Dataset | A(orz, 4/4, $p$=0.7) | | | | B(dm, 4/4, $p$=1.0) | | | | C(dm, 2/4, $p$=1.0) | | | | D(dm, 4/4, $p$=0.7) | | | |
|---|---|---|---|---|---|---|---|---|---|---|---|---|---|---|---|---|
| | Max | Maj | P@1 | Avg | Max | Maj | P@1 | Avg | Max | Maj | P@1 | Avg | Max | Maj | P@1 | Avg |
| AIME25 | 63% | 53% | 35% | 35% | 50% | 40% | 28% | 27% | **66%** | 40% | 30% | 29% | 56% | 36% | 31% | 30% |
| AIME24 | **76%** | 60% | 42% | 41% | **76%** | 63% | 48% | 48% | 66% | 63% | 42% | 42% | **76%** | 50% | 43% | 43% |
| HMMT Feb. 25 | 53% | 20% | 19% | 19% | **60%** | 33% | 25% | 25% | 50% | 23% | 22% | 22% | 50% | 33% | 22% | 22% |
| HMMT Feb. 24 | 53% | 33% | 25% | 24% | 53% | 33% | 25% | 24% | 46% | 26% | 22% | 22% | 50% | 40% | 26% | 26% |
| CMIMC | 60% | 53% | 34% | 34% | **63%** | 33% | 30% | 30% | **63%** | 33% | 30% | 30% | 56% | 33% | 31% | 31% |

## 4.4 Analysis of Training Dynamics

Figure 4 presents key training dynamics, offering insights into the learning process and the Agent RL Scaling Law across different experimental settings. The evolution of code-related metrics is particularly revealing. Code Proportion, representing spontaneous code usage frequency, consistently shows an initial dip followed by a significant increase for TIR-enabled models, indicating that the agent learns the utility of the tool over time after overcoming initial generation challenges. Concurrently, the Code in Correct metric, tracking correct answers involving code, rises sharply in conjunction with the Raw Reward Avg, empirically linking effective, learned tool use directly to task success.

Response Length generally increases with training, especially for larger models, correlating with the inclusion of code and outputs, though this trend does not perfectly mirror reward improvements across all settings. The reward curves clearly confirm that enabling tool interaction leads to superior

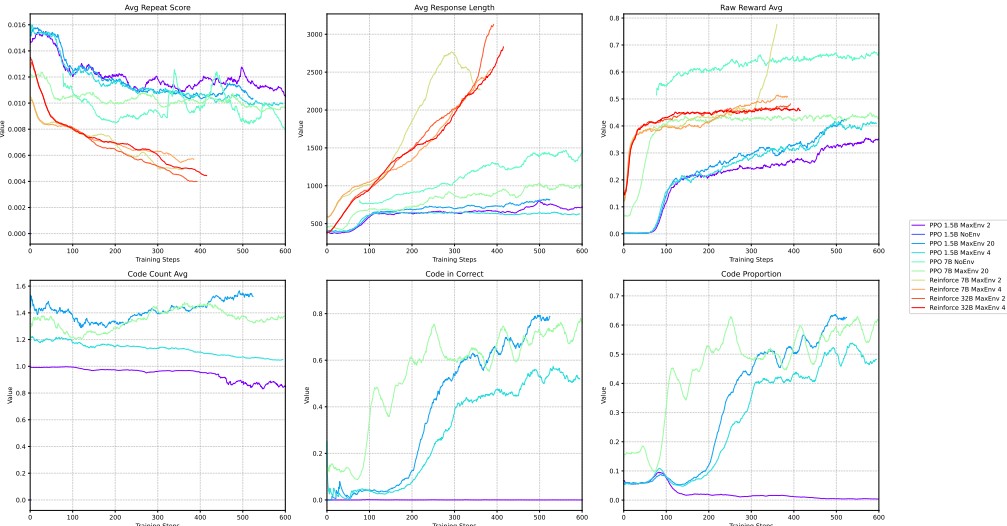

Figure 4: Training dynamics comparing ZeroTIR and ZeroRL in different model and RL algorithm.

performance compared to non-tool baselines, and further illustrate positive scaling with both increased maximum allowed interactions $N_{max}$ and larger model parameter counts.

Interestingly, while ablations show performance benefits from higher $N_{max}$ values, the Code Count Avg across different settings often stabilizes between 1 and 2 calls per response. This suggests that although a larger interaction budget is available and beneficial, the agents predominantly learn strategies involving few interactions. Combining these observations, we conclude that for base models learning via ZeroTIR, early attempts at multiple interactions might yield poor rewards due to lower code quality, leading the agent to converge towards highly effective single-call strategies. Indeed, our analysis indicates that over 90% of correct responses involving code utilize only a single code execution, with fewer than 10% employing two or more calls, even when permitted.

### 4.5 Joint scaling of training and inference-time interactions

Figure 5 shows the accuracy obtained when we train agents with 2, 4 or 8 interactions per problem and then test them with 1–16 interactions. On every benchmark, allowing more interactions at inference time yields higher scores, but the improvement becomes progressively smaller once the budget reaches 8–16 steps. The curves also reveal a clear interaction between the two budgets: models that were trained with many interactions profit the most from a generous inference budget, yet they can be slightly worse than the low-budget model when only a single step is permitted at test time. In other words, heavy reliance on multi-step reasoning during training can hurt in the one-shot regime, while paying off strongly once at least four inference steps are available.

The extent of these effects varies across datasets. On the harder AIME24/25 sets the gap between the 8-interaction model and the 2-interaction model widens to more than ten percentage points when sixteen inference steps are allowed, whereas on HMMT24/25 the corresponding gap is around seven points. Taken together, the results suggest that training and deployment budgets should be aligned: if an application can only afford one or two reasoning steps it is safer to train with the same constraint, but if a larger inference budget is feasible, giving the model ample interaction capacity during training unlocks noticeably better final accuracy.

## 5 Conclusion

This work investigated the autonomous acquisition of tool use, specifically spontaneous Python code execution for mathematical reasoning, by base Large Language Models trained via outcome-based Reinforcement Learning, ZeroTIR. Our central contribution is the identification and empirical characterization of Agent RL Scaling Law governing this learning process. We showed that training progression leads to predictable dynamics in tool usage frequency, code quality, response length, and task accuracy. Key findings include the positive scaling of performance with both model size and

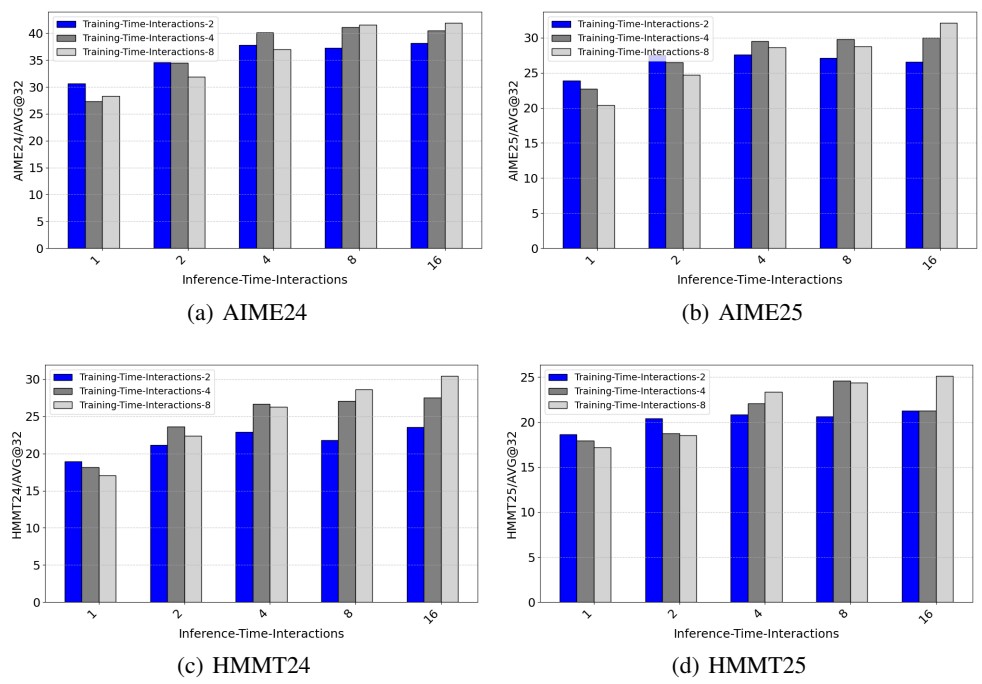

(a) AIME24      (b) AIME25

(c) HMMT24      (d) HMMT25

Figure 5: Training-Time-Interactions and Inference-Time-Interacrtions. We use average@32 to compute the score for AIME24, AIME25, HMMT24 and HMMT25. We use 500-step checkpoint for different training-time-interactions to perform the fair evaluation.

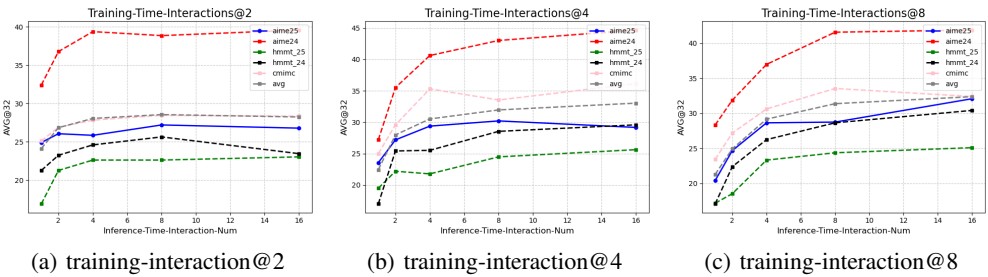

(a) training-interaction@2    (b) training-interaction@4    (c) training-interaction@8

Figure 6: Training time interactions and performance (AVG@32). Increasing the inference interaction number from 2 to 4 to 8 improves all datasets. AIME24 shows the largest and nearly monotonic gains, and AIME25 is second. HMMT 24 and 25 rise steadily. CMIMC peaks around 4 interactions and then slightly declines but remains above 2. The overall average flattens near 8 interactions, so using 8 interactions gives a good balance between accuracy and cost.

the allowed budget for environment interactions, alongside insights into algorithm efficiency where Reinforce++ showed faster convergence than PPO in our experiments, and the deepmath dataset yielded strong results. Crucially, we revealed typical patterns of Tool-Integrated Reasoning usage that emerge during ZeroRL training. Our analysis of training dynamics and interaction counts suggests that while increased interaction potential improves results, models often converge to strategies favoring fewer, high-utility code calls, with a majority of successful tool-using solutions employing only a single execution. Due to resource limitations, this work primarily focused on empirically demonstrating the existence and qualitative nature of these scaling law. A rigorous quantitative analysis to determine the precise mathematical form of these relationships remains an important direction for future work. Further research should also explore removing constraints on interaction counts entirely and investigating performance in even more complex agentic environments. Overall, our findings advance the understanding of autonomous tool learning in Agent RL and provide a reproducible framework for studying these critical scaling effects.

## Acknowledgments

This work was supported by National Natural Science Foundation of China (No.62576109, 62072112, 12471280, 12101241) and a grant from the Shanghai Municipal Education Commission (No.2024AI01002).

We also thank Zhong-Zhi Li and Jian Hu for valuable advice and discussions.

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

# A Statistical Evidence of the Scaling Relationship

We sample $\sim$500 training steps batch data and compute the association between code/tool-usage rate and task accuracy. We observe a strong positive monotonic correlation with Spearman $\rho = 0.686$ and extremely small $p$-values. A log–log linear fit between effective training steps and accuracy yields $R^2 = 0.93$, indicating a predictable trend as the training budget grows. We also observe a characteristic trajectory from a non-math, non-TIR base: the fraction of responses using code initially dips ($0.142 \rightarrow 0.002$) before rising to a high plateau ($> 0.98$), alongside the accuracy increase after the inflection.

Table 4: Summary statistics linking tool usage and accuracy.

| Statistic | Value | Note |
|---|---|---|
| Spearman $\rho$ (usage vs. acc.) | 0.686 | strong positive association |
| Log–log fit $R^2$ (steps vs. acc.) | 0.93 | high goodness of fit |
| Early usage dip $\rightarrow$ plateau | $0.142 \rightarrow 0.002 \rightarrow > 0.98$ | phase transition pattern |

**Base-model rationale.** Effective tool use is a prerequisite for outcome-only training. We therefore adopt a base with better tool-utilization priors, which empirically reduces tool over-calling and hallucination and stabilizes the RL signal.

# B Tool-Call Efficiency and "Laziness"

We analyze the number of tool invocations per example. Incorrect answers average 2.7 calls versus 1.9 for correct answers. As training progresses, tool calls become more efficient: $> 90\%$ of correct code-using cases require a single call. Increasing the maximum allowed interactions $N_{\max}$ improves accuracy with diminishing returns beyond $\approx 4$; competent models typically solve with 1–2 calls.

Table 5: Tool-call efficiency analysis. Lower calls for correct solutions indicate non-brute-force behavior.

| Metric | Value | Observation |
|---|---|---|
| Avg. calls (incorrect) | 2.7 | higher than correct |
| Avg. calls (correct) | 1.9 | lower, more efficient |
| Single-call share (correct, code-using) | $> 90\%$ | efficiency improves |
| Benefit of $N_{\max}$ | diminishing beyond $\sim 4$ | accuracy gains taper off |

# C Extended ZeroTIR Training and Validation Results

During the extended experiment phase, we reran the ZeroTIR training process up to 1,500 steps and recorded the average@32 accuracy on all three evaluation benchmarks (AIME24, AIME25, and HMMT25). The complete trend of training-step progression and corresponding validation accuracy is summarized in Figure 8.

**Key observations.** Accuracy continues to improve or remain stable up to 1,500 steps, with no sign of degradation or collapse. Both AIME24 and HMMT25 exhibit noticeable late-stage gains, indicating that the agent benefits from additional computation without overfitting to earlier exploration patterns. These results further support the stability and scalability of the ZeroTIR framework under long-horizon training.

# D Cross-Dataset Generalization and Comparative Trends

To complement the single-run results presented above, we additionally tracked the performance of ZeroTIR across multiple datasets, iterations, and baselines. Figure 7 visualizes the extended evaluation curve covering AIME24, AIME25, and the averaged five-dataset setting, compared against a non-thinking baseline (Claude-Optus4).

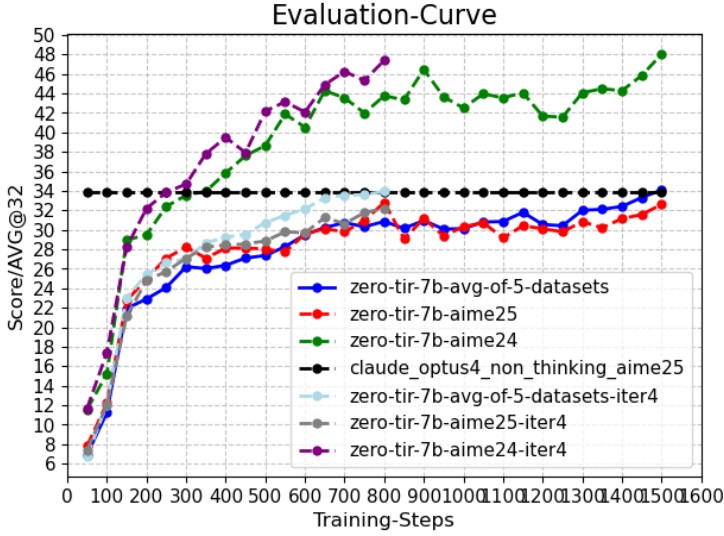

Figure 7: Cross-dataset evaluation under prolonged training. The ZeroTIR-7B series demonstrates stable and monotonic improvement across all datasets up to 1,500 steps, surpassing the non-thinking baseline and showing no overfitting or collapse.

**Observations.** The averaged five-dataset curve (blue) provides a robust indicator of global generalization, with accuracy continuing to rise smoothly through extended optimization. Both single-task and averaged variants of ZeroTIR consistently outperform the static baseline across the full training horizon. These results validate that ZeroTIR's improvement is not confined to a specific benchmark but reflects a scalable and transferable training dynamic across diverse reasoning domains.

# E    Limitations

The strongest results here are math-centric. While the infrastructure is general and decoupled, broader evaluations across non-math tasks, alternative tools, and efficiency-oriented rewards are left for future extensions to assess the scope and boundary conditions of the observed scaling behaviors.

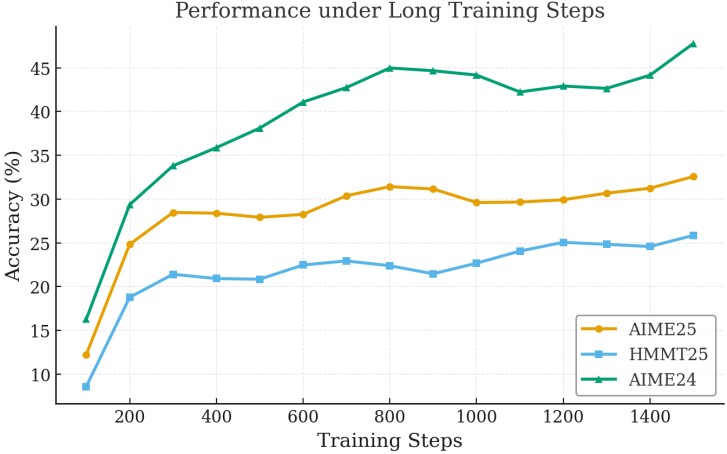

Figure 8: Smoothed validation accuracy curves for AIME24, AIME25, and HMMT25 across 1,500 training steps. Accuracy continues to improve or stabilize, showing no collapse and confirming late-stage gains.

