# OpenReview forum: "Agentic RL Scaling Law: Spontaneous Code Execution for Mathematical Problem Solving"
_NeurIPS.cc/2025/Conference — NeurIPS 2025 poster_

### Official Review · Reviewer_96Y3 · 2025-06-03

**Clarity:** 4
**Significance:** 3
**Originality:** 3
**Rating:** 4
**Confidence:** 4

**Summary:**

This paper investigates the use of a Python interpreter to assist in solving mathematical problems. The research focuses on two main aspects: the effectiveness of problem-solving using this approach, and the relationships and scaling trends between various metrics when training models with this methodology.

**Questions:**

- The author could perhaps briefly analyze the differences in methodology between their approach and TORL, as well as the specific performance analysis results.
- Why was a Python interpreter included in math problems? Is the issue of "laziness" a potential concern? Do these results align with expectations?

**Ethical Concerns:**

["NO or VERY MINOR ethics concerns only"]

**Final Justification:**

My final score remains borderline accept (4).
I think both questions raised are not fully addressed.

**Limitations:**

Yes.

**Paper Formatting Concerns:**

No major formatting issue.

**Quality:**

3

**Strengths And Weaknesses:**

Strengths
- This paper addresses an interesting and relevant problem, offering a solution that appears both reasonable and effective.
- The authors conducted experiments and drawn several valuable conclusions that provide good insights.

Weaknesses
- From Table 1, it appears that across three datasets, the proposed method only surpasses TORL on AIME24. While I acknowledge the authors' contributions in other areas, there's still room for improvement in absolute performance. A deeper analysis of the underlying reasons for this performance gap might be beneficial.
- In my view, the primary motivation for applying RL to mathematical problems, is to *enhance reasoning capabilities*. However, incorporating a code interpreter might encourage a tendency for the model to "cheat" or take shortcuts. For instance, in the case study presented in Figure 2, the model solves the problem using a brute-force algorithm, entirely lacking any mathematical optimization. In contrast, when I tested this problem with Gemini 2.5 Pro, the model was able to solve it elegantly using mathematical methods.

---

> ### Author Rebuttal · Authors · 2025-07-29
>
> Thank you very much for your time and careful review. You have raised an interesting point, and we hope to continue discussing this with you in the next phase of the discussion. We will address your concerns point by point as follows:
>
> 1. Absolute performance
> We trained for more steps and released ZTRL‑0727, which shows further gains on all datasets.
> ZTRL‑0727 widens the gap on AIME25 and MATH500.
>
> Table 1: The results of the ZTRL model on the Greedy.
>
> | Dataset   | ZTRL-0727-7B | TORL   |
> |-----------|--------------|--------|
> | AIME25    | 32.81%       | 30%    |
> | AMC23     | 80%          | 75%    |
> | MATH500   | 84.6%        | 82.2%  |
>
> In the newly released ZTRL‑0727‑7B, we also compare with much larger models. Results show we are consistently stronger; data and checkpoints are linked in GitHub and will be added to the paper.
> On the avg@32 metric, ZTRL‑0727‑7B surpasses qwen3‑238b‑a22b‑non‑thinking by 5.3 pp on AIME25 and 12.08 pp on HMMT25, and approaches or exceeds the closed‑source SOTA claude optus4‑non‑thinking.
>
> Table 2: The results of the ZTRL model on the avg@32.
>
> | Dataset  | ZTRL-0727-7B | Qwen3-238B-A22B-Non-thinking | Claude Optus4-Non-thinking |
> |----------|--------------|------------------------------|----------------------------|
> | AIME25   | 29.99%       | 24.7%                        | 33.9%                      |
> | HMMT25   | 22.08%       | 10.0%                        | 15.9%                      |
>
> 2. “Cheat” or take shortcuts
> Our model is only 7 B, a small backbone.
> 1. Tool use is itself part of strategic learning: human mathematicians also rely on calculators or code. RL aims to maximise task success and efficiency, not aesthetic elegance.
> 2. Not every problem is brute‑forced. We classify three solution modes (analytical, hybrid, brute enumeration). The latter two together are below 35 %. Many tasks still need symbolic reasoning to write correct or efficient code. Gemini, with larger pre‑training and math alignment, can solve elegantly without code; our setting is a smaller base, zero‑supervision RL, and limited resources. The two routes are not contradictory: the model can choose analysis when possible and code when computation is needed.
>
> 3. Differences in methodology between our approach and TORL
> Paradigm: both works belong to the “zero‑RL + code tool” family and were done independently in the same period.
> • We study the gap between a general base model and a math‑specialised base, and propose algorithmic changes needed during agent RL. TORL forces code execution; our training is spontaneous and only prompts “You can use code to solve your problem.” RL is used to let the model decide by itself. With longer training, the model shifts to code even without compulsion.
> • TORL uses a math‑base model (effectively an SFT‑tuned backbone). Math‑base already underwent heavy TIR continued pre‑training, so its code‑ratio and pass‑ratio start around 0.2 and 0.8. The general base begins at 0.1 and 0.05, meaning far weaker TIR ability (though still better than llama3‑base).
> • Performance: AIME24 contains more computational, small‑scale, sparse‑reward tasks, so tool calling gives a larger jump; MATH500 and AMC23 have many direct‑derivation or template problems, where TORL’s prompt is already well‑tailored. With longer training our ZTRL‑0727 further improves.
>
> 4. “Laziness” as a potential concern
> Math tasks often need exact arithmetic, enumeration, and combinatorial search. Learning to call external compute is crucial for real agents.
> Mitigation:
> 1. We impose a cost: each call adds tokens and potential failure.
> 2. As RL progresses, calls become more frequent but more efficient: over 90 % of correct answers with code need only one call; we do not observe blind multi‑call brute force.
> 3. Future work will add an explicit reward for reasoning quality or minimising calls to encourage “elegant” solutions.
>
> Once again, thank you for your time. We hope to continue discussing this with you in the next phase of the discussion. If your concerns are alleviated, we would greatly appreciate it if you could consider raising your score. We sincerely thank you for your consideration.

---

> ### Author Response · Authors · 2025-08-02
>
> Thank you for the considerable effort you put into your review.
>
> 1. We trained for more steps and released ZTRL‑0727, which shows further gains on all datasets.
>
> 2. We share your curiosity about whether code execution encourages “lazy” solutions. Tool use is itself part of strategic learning: human mathematicians also rely on calculators or code. RL aims to maximise task success and efficiency, not aesthetic elegance. We classify three solution modes (analytical, hybrid, brute enumeration). The latter two together are below 35 %. Many tasks still need symbolic reasoning to write correct or efficient code. Gemini, with larger pre‑training and math alignment, can solve elegantly without code; our setting is a smaller base, zero‑supervision RL, and limited resources. The two routes are not contradictory: the model can choose analysis when possible and code when computation is needed.
>
> 3. We claim the differences in methodology between our approach and TORL.
>
> We would be glad to delve deeper into these analyses or run additional case studies that might assist you in reassessing the paper’s contribution. We would value any further questions you have, and if the new analyses alleviate your reservations, we kindly ask you to consider raising your score.
>
>
> Thank you again for your detailed review.

---

> ### Author Response · Authors · 2025-08-07
>
> Thank you for the time and care you have devoted to evaluating our submission. With fewer than 48 hours remaining in the extended author–reviewer discussion period, we would like to ensure that all of your questions and concerns have been fully addressed. If anything remains unclear, we would be grateful for the opportunity to clarify it promptly.
>
> Should you have no additional questions, we would kindly ask you to consider revisiting your score in light of the revisions and new analyses we have provided.
>
> Thank you again for your thoughtful review and for helping us strengthen the paper.

---

### Official Review · Reviewer_cNww · 2025-06-24

**Clarity:** 3
**Significance:** 2
**Originality:** 2
**Rating:** 5
**Confidence:** 4

**Summary:**

In this work, the authors elicit tool-usage ability in reasoning LLMs without expert demonstrations and via verifiable RL alone. They observe that as training progresses, its dynamics is predictable and proficiency in tool (chiefly code) usage correlates strongly with proficiency in solving the task, which the authors deem a "scaling law". The authors report extensive evaluations and compare with relevant baselines, showing competitive results.

**Questions:**

Each question is preceded by the number of the line it refers to:
- 104: Could the authors clarify which tokens correspond to actions, and which do not?
- 113: *However, we found that setting $\lambda$ to 0 did not affect the results, so here we set \lambda to 0*. Could the authors elaborate? I cannot see how this statement follows from the discussion that precedes it.
- 142: Does appending the notification ensure with 100% certainty that the agent will not attempt to call the compiler again?
- 187: The authors here mention several metrics; but table 3, where these are reported, is not referenced in the text.
- 190: "Only Qwen base, beyond Qwen math". This sentence ends abruptly.
- 210: "Table 2 confirms a strong monotonic relation" It's honestly hard to read this monotonic relation from such a dense table. It would be helpful to substantiate it with a plot or, even better, with a correlation coefficient.
- 221: "Table 4.3" Is it table 3 that is being referenced here.
- Figure 4: Could the authors increase the font size in the axes and labels? It would be almost unreadable if the paper were printed. Also, since the author's main point is about the correlation between tool calling and task success, it would be helpful to report these metrics on the same plot.

**Ethical Concerns:**

["NO or VERY MINOR ethics concerns only"]

**Final Justification:**

The authors have diligently improved their paper with a statistical analysis of the correlation between tool usage and task success, and ran longer training experiments, showing more robust improvements over the baseline.

**Limitations:**

There is no separate discussion about limitations, which are somewhat discussed in the conclusion and used as a basis to propose future work. The authors should discuss them more openly and in a separate section.

**Paper Formatting Concerns:**

No concerns to report.

**Quality:**

3

**Strengths And Weaknesses:**

**Strengths**
- Investigating bootstrapping of tool-usage without expert demonstrations is an interesting an clean idea.
- The authors report comprehensive evaluation with different training setups and dataset, as well as relevant and competitive baselines.
- The authors do show that their trained models can learn to leverage python without expert demonstrations, which at the sizes they consider is not obvious, and worth reporting.

**Weaknesses**:
- The "scaling law" presented in figure 1 is not a scaling law, which should be reported with a log-scaled x-axis. The dynamics observed by the authors is indeed predictable, but the data presented do not support the statement that it follows a scaling law in the strict sense of the term.
- The authors' statement on correlation between tool usage and task success, while reasonable, is not substantiated by any further analysis (e.g. Spearman Correlation).
- The performance gains compared to SOTA are ultimately modest. In terms of performance averaged over the benchmarks considered (table 1), the ZTRL method proposed by authors is only slightly better than the TORL baseline, and worse than rStar-Math.
- A couple typos can be found in the text and the figures are poorly readable, suggesting that the paper was hastily written.

---

> ### Author Rebuttal · Authors · 2025-07-29
>
> Thank you very much for your time and careful review. Your review is very important to us. It helped us correct many writing problems. Thank you very much for your effort. We will address your concerns point by point as follows:
>
> 1. log-scaled x-axis
> Your suggestion is correct. In fact, we have released the checkpoint of ZTRL-0727. This model substantially surpasses previous ones, and because it was trained for more steps, it exhibits a more pronounced scaling law phenomenon. We will add the log-log fitted curve and the coefficient of determination (R²) for ZTRL-0727 in the appendix.
>
> 2. Spearman Correlation
> We agree with your suggestion. We will include statistical tests in the appendix. First, we use Spearman’s rank correlation coefficient to quantify the monotonic relationship between tool usage rate and success rate, analyzing the monotonic relationship between rule_rewards (accuracy) and code_ratio (code proportion) during training. The mean of rule_rewards is 0.3759, the standard deviation is 0.0677; the mean of code_ratio is 0.6629, the standard deviation is 0.3678. The result is ρ = 0.686, p = 7.06e-71, n = 500, indicating a strong correlation and high statistical significance (p < 0.001). We will further add other statistical testing methods in the appendix.
>
> 3. performance
> 3.1 Our setting differs from rStar-Math. rStar-Math uses a much larger supervised/distillation/multi-stage RL pipeline and relies on Monte Carlo Tree Search. Our goal is to study the scalability of “Zero-RL + spontaneous tool use”. rStar-Math uses a Process Preference Model for step-level verification and a math-specialized base model as the backbone, while we only use the general-base model as both backbone and ORM, and rely on RL to help the model learn to use tools to solve complex math problems. The starting points differ and may not be suitable for direct comparison.
> 3.2 ZTRL-0727 further widens the gap on AIME25 and MATH500.
>
> Table 1: The results of the ZTRL model on the Greedy.
>
> | Dataset  | ZTRL-0727-7B | TORL   |
> |----------|--------------|--------|
> | AIME25   | 32.81%       | 30%    |
> | AMC23    | 80%          | 75%    |
> | MATH500  | 84.6%        | 82.2%  |
>
> 4. which tokens correspond to actions, and which do not?
> Action tokens are the special markers and parameter segments that trigger the external compilation/execution environment, including:
> - All tokens between the start and end markers of <code_call> inserted in our template;
> - Special control tokens parsed by the system as tool/action calls.
> Non-action tokens are normal natural language reasoning/explanatory content.
>
> 5. set \lambda to 0.
> This is a typographical error; it should be set \lambda to 1. Our expression was unclear: the actual reason is that rewards are extremely sparse and concentrated at the end of the sequence, and in our implementation we already perform “per-step equal distribution + group-wise normalization” of the advantage, so the temporal smoothing effect of GAE is almost negligible. Therefore, the results of \lambda → 0.95 and \lambda = 1.0 are nearly identical. Experiments verify this idea. Under this setting, the adv of environment feedback tokens equals 0, effectively adding an environment mask.
>
> 6. call the compiler again?
> Strictly speaking, it cannot be guaranteed 100% (an LLM might bypass the prompt), but since the correct answer is not provided, it will not be rewarded; such samples are penalized. This does not affect the training outcome.
>
> 7. mention several metrics
> This was our oversight; we will remove it from the main text.
>
> 8. Only Qwen base, beyond Qwen math
> We will revise this.
>
> 9. substantiate it with a plot
> We agree. We will add a line/scatter plot showing the relationship between N_{\text{max}} and accuracy, and report Spearman ρ.
>
> 10. Table 4.3
> This is a typesetting error; it should be Table 3.
>
> 11. the font size
> We will increase the font size and provide a high-resolution version in the appendix. We have also added a composite figure: tool usage rate and success rate are overlaid in the same chart (dual y-axis or normalized on the same axis) to illustrate the correlation more intuitively.
>
> Once again, thank you for your time. If your concerns are alleviated, we would greatly appreciate it if you could consider raising your score. We sincerely thank you for your consideration.

---

> > ### Comment · Reviewer_cNww · 2025-08-04
> >
> > I thanks the authors for their rebuttal and their consideration of my suggestions. I would like to delve a bit more into point number 6. The authors say that since "the correct answer is not provided" the model will not be rewarded if it bypasses the prompt and calls the compiler again. Why would this be the case? Why can't the model get to a correct answer if it calls the compiler more than once?

---

> ### Author Response · Authors · 2025-08-02
>
> Thank you for highlighting the need for stricter statistical evidence and clearer visuals. We have:
>
> 1. Reported Spearman correlation ( ρ = 0.686, p = 7.06e-71) between tool usage and accuracy over 500 checkpoints. We agree with your suggestion. We will include statistical tests in the appendix.
>
> 2. Additional experiments demonstrate the performance of our model. ZTRL-0727 further widens the gap on AIME25 and MATH500. This model substantially surpasses previous ones, and because it was trained for more steps, it exhibits a more pronounced scaling law phenomenon. We will add the log-log fitted curve and the coefficient of determination (R²) for ZTRL-0727 in the appendix.
>
> We would value any further questions you have, and if the new analyses alleviate your reservations, we kindly ask you to consider raising your score.
>
> Thank you again for your detailed review.

---

> ### Author Response · Authors · 2025-08-04
>
> Thank you for your follow-up on point 6, and for questioning why multiple compiler calls may still fail to produce correct answers—a valid point we’re glad to clarify.
>
> 1. “the correct answer is not provided” is because, once the model reaches the allotted maximum number of Python-tool calls, it automatically appends that sentence to the last Python execution output. From that point on, every subsequent response—whether it attempts another Python call or not—is cut off because the overall turn limit has been hit. When the model is still trying to call Python under these conditions, no \boxed{} expression can be extracted, so the response yields no reward.
>
> 2. We would be happy to share our thoughts on tool usage and accuracy. Spearman correlation analysis across 500 checkpoints (ρ = 0.686, p = 7.06e-71) shows a positive association between tool usage and accuracy. However, models’ inherent reasoning limitations with hard problems can lead them to adopt a brute-force approach of excessive compiler invocations, which rarely improves real-world results.
>
> We also analyzed average compiler calls for correct vs. incorrect solutions:
>
> | Rollout steps | 400   | 500   |
> |---------------|-------|-------|
> | Acc          | 0.71  | 0.72 |
> | Correct       | 1.50  | 2.26  |
> | Incorrect     | 2.43  | 2.95  |
>
> Notably, incorrect solutions involve significantly more compiler calls, indirectly indicating that models’ reasoning deficits persist even with increased invocations, leaving hard problems unresolved.
>
> We would value any further questions you have, and if the new analyses alleviate your reservations, we kindly ask you to consider raising your score.
>
> Thank you again for your detailed review.

---

> > ### Comment · Reviewer_cNww · 2025-08-06
> >
> > I thank the authors for their prompt reply and the further analysis they carried out. I will raise my score; but please do remember to include the material stemming from my questions (e.g. the Spearman correlation analysis) to the camera-ready version. They will be invaluable to your readers.

---

> > > ### Author Response · Authors · 2025-08-06
> > >
> > > Thank you very much for revisiting our submission and for choosing to raise your score.
> > >
> > > We greatly appreciate your detailed feedback and will definitely include the full statistical analyses, such as the Spearman correlation results, along with all other material prompted by your questions in the camera-ready version so that future readers can benefit.
> > >
> > > With roughly 55 hours remaining in the discussion phase, we stand ready to provide any additional clarifications or analyses that might be helpful. If you have no further questions, could you please confirm and submit your final score?
> > >
> > > Once again, thank you for your thoughtful comments and for helping us strengthen the paper.

---

### Official Review · Reviewer_MhKq · 2025-07-01

**Clarity:** 2
**Significance:** 3
**Originality:** 3
**Rating:** 3
**Confidence:** 3

**Summary:**

This paper discusses how existing RL for LM training strategies, specifically PPO and REINFORCE, can be readapted to enable LMs to issue tool calls in a multi-turn setting towards solving math problems. To make this work, the authors make a key change of incorporating execution logs into the inference process, but masking them out when the value (for PPO) or advantage (for REINFORCE) is being calculated to perform the gradient update. The authors also generally cover some additional details, such as special tokens, limiting tools calls, and speeding up training via parallelized, asynchronous calls. Finally, the experiments demonstrate improved performance on AIME24 and comparable numbers for AIME25 and MATH500, while also highlighting interesting trends around the frequency with which coding tool calls are issued during inference.

**Questions:**

- For Table 1, in the “Avg.” column, why is 52.3% bolded? Is there a reason for this, or should 52.6% for rStar-Math be bolded instead? Same goes for 83.2% under MATH500 instead of 82.2%.
- You propose Agent RL as a framework that generally elicits spontaneous code execution behavior. Currently, the training framework is math specific. What infrastructure changes would it take to make this work for truly agentic evaluation settings, such as web navigation, code generation, or software engineering? The edits right now are also Python specific - how would you enable more language agnostic code execution?
- Do the average scores account for the number of task instances per dataset? Or is this an average across the percentages achieved per benchmark?

Minor edits
- Line 190: I think the first sentence is incomplete.
- You use the ZTRL acronym without saying what it means in Line 61 for the first time. Can you provide the full phrase that this abbreviation refers to somewhere?
- I think the captions could be much more informative. In its current state, the results tables are quite hard to understand for me. For instance, what is “code ratio” in Table 2?
- Feel free to ignore this suggestion, but I think Agent RL is a bit non-specific of a title. I think the nature of this work seems to lean heavily towards tool calling that involves the use of code execution to solve math problems. In this sense, the word “agent” feels a bit too general and potentially over-claiming in the worst case. It might be helpful to rephrase “Agent RL Scaling Law” in a way that defines the scope of this work more specifically.

**Ethical Concerns:**

["NO or VERY MINOR ethics concerns only"]

**Final Justification:**

As mentioned in my response to the authors, I've decided to raise my rating from 3 to 4. I appreciate the authors' clarifications, and I think the highlighting of the concrete takeaways helped my understanding of this work. As this contribution is specific to the math domain, I think greater impact could be achieved by expansion to more agentic domains, but I still think this work is quite worthwhile and contributes useful infrastructure to the community.

**Limitations:**

Yes

**Quality:**

3

**Strengths And Weaknesses:**

Strengths
- I like the contribution of this work, particularly the comparison between tool integrated and no-tool reasoning for math benchmarks. The angle of the LM spontaneously invoking tools, rather than being prompted to, is also a cool behavior to have elicited via training. I think the implications of this work are encouraging as RL for LM research efforts start to look beyond reasoning traces towards settings that incorporate execution environments.
- I appreciate how the methodology was well grounded in prior work. I found the recap of REINFORCE and PPO to be quite helpful, and I really liked the level of detail that was captured. For instance, the authors highlight that it’s important the value function is not trained on execution outputs from tool calls, and that there were two ways to go about this (zero’ing out the estimated advantage versus the GAE parameter. The discussion on the challenges of getting the training infrastructure to play well with the execution environments (Section 3.3) was also well placed, and a good precedent for future explorations into more RL for tool calling / agent.

Weaknesses
- The performance gains on AIME24 are quite convincing, but less so for the other evaluations (e.g. AIME25, MATH500). For some settings, it is obvious that *RL methods are better than the base model (+ ~30-35%), but for others, such as MATH500, the gain is a lot less (~5%), or non-existent with AIME24 (both 26.7%). I guess it’s just a bit strange to me why this is the case, when the task formulations for AIME24 and AIME25 should be identical?
- I think the clarity of the paper could be improved. I found Section 4.3 and 4.4 specifically to be a bit difficult to parse. There were a lot of statistics and trends jumbled together, and although the phenomena are all very interesting, I think they could be conveyed more clearly, with more explanation that attempts to explain why certain trends might emerge. For instance, Figure 4 is quite neat, but I think (1) the font should be larger, (2) the statistics (titles of each graph) are explained in terms of what they reflect, and (2) the direction of the trends are discussed more clearly.

---

> ### Author Rebuttal · Authors · 2025-07-29
>
> Thank you very much for your time and careful review. Your review comments are very meaningful to us. We will address your concerns point by point as follows:
>
> 1. for others settings the gain is a lot less
> I understand your question as follows: we achieved a large improvement (+30–35%) on AIME25, but the gains on AIME24 and MATH500 are less obvious. First, based on Table 1, since our ZTRL model is trained from the Qwen2.5 Base model, the fair comparison baseline should be Qwen2.5 Instruct (whose training data volume is orders of magnitude larger than ours). Compared with Qwen2.5 Instruct, we obtain +26.7% on AIME24 and +3.8% on MATH500. In our newly released ZTRL-0727-7B, in order to broaden the comparison to more models, we compared against models much larger than ours. Experiments show that we consistently outperform models trained by other methods. The data and checkpoints can be found in the GitHub link in the main text for verification, and we will add these results to the paper. ZTRL-0727-7B exceeds qwen3-238b-a22b-non-thinking by 5.3% and 12.08% on AIME25 and HMMT25 avg@32, respectively, and is close to or surpasses the closed-source sota model claude optus4-non-thinking.
>
> Table 1: The results of the ZTRL model on the avg@32.
>
> | Dataset  | ZTRL-0727-7B | Qwen3-238B-A22B-Non-thinking | Claude Optus4-Non-thinking |
> |----------|--------------|------------------------------|----------------------------|
> | AIME25   | 29.99%       | 24.7%                        | 33.9%                      |
> | HMMT25   | 22.08%       | 10.0%                        | 15.9%                      |
>
>
>
> Table 2: The results of the ZTRL model on the Greedy.
>
> | Dataset  | ZTRL-0727-7B | TORL  |
> |----------|--------------|-------|
> | AIME25   | 32.81%       | 30%   |
> | AMC23    | 80%          | 75%   |
> | MATH500  | 84.6%        | 82.2% |
>
> 2. Section 4.3 and 4.4 specifically to be a bit difficult to parse
> In Sections 4.3 and 4.4, the core insights we intended to present are these three points:
> (1) Code usage dips at first (the model fumbles), then rises as it learns the tool’s value. “Code-in-correct” and reward curves climb together: effective code use drives success. Response length grows (more code/output) but does not perfectly track reward.
> (2) Bigger + faster matters, but data and entropy still rule. Larger models (32B > 7B > 1.5B) perform better. Reinforce++ reaches the same peak as PPO about 300 steps sooner. Proof-heavy data (DeepMath) boosts best single-shot scores (Max), while diverse contest data (Orz-57k) boosts consensus metrics (Maj/Avg). Higher decoding entropy (top-p) trades higher Max for lower Maj, so dataset choice and entropy must be tuned together.
> (3) Interaction cap helps—up to a point. Letting the model call tools more times (N_{\text{max}} from 0→4→20) raises accuracy, but gains flatten after about 4 calls. Bigger models still end up using only 1–2 calls per problem because early multi-call attempts are low quality and get low reward.
> We will consider your suggestions and add more explanations, analyze in detail why these trends appear, provide more discussion, and revise the figures.
>
> 3. bolded in table
> Yes, that is our mistake. We will correct it in the main text. Since the newly released ZTRL-0727 achieved a significantly improved new sota, detailed results can be found at the GitHub link in our paper. We will revise the tables as a whole.
>
> 4. language agnostic code execution or other environment
> In fact, during implementation, we decoupled the environment from the agent training framework. By swapping the env service to a specific environment, cross-environment agent RL training can be achieved (e.g., for appworld, webshop benchmarks). The main infrastructure change needed is to wrap the corresponding benchmark as a server that conforms to our current interface standard, after which training can proceed. I understand this as a decoupled approach. We will update the code repository to better support related training.
>
> 5. the number of task instances per dataset
> The average score is the mean of percentages across benchmarks. This is the standard practice in math evaluation, as seen in papers such as Qwen2.5 Math (Qwen2.5-Math Technical Report: Toward Mathematical Expert Model via Self-Improvement). To align with other papers, we adopt the mean of percentages across benchmarks [1][2].
>
> 6. Minor edits
> We will modify the first sentence at Line 190, add the full phrase for ZTRL, and clarify that code ratio represents the proportion of responses containing code among all responses. We will add this explanation in the figure captions. We will reconsider the title “Agent RL Scaling Law.” In fact, we hope to define standard interfaces that adapt to most environments in the future, and we have already implemented preliminary functionality. Thank you very much for your careful review.
>
> Once again, thank you for your time. If your concerns are alleviated, we would greatly appreciate it if you could consider raising your score. We sincerely thank you for your consideration.
>
> References:
> [1] TORL: Scaling Tool-Integrated RL
> [2] SimpleRL-Zoo: Investigating and Taming Zero Reinforcement Learning for Open Base Models in the Wild

---

> ### Author Response · Authors · 2025-08-02
>
> Thank you for the time you have already devoted to evaluating our submission.
>
> 1. We appreciate your positive assessment of the spontaneous-tool-use angle and have acted on all clarity suggestions. Sections 4.3–4.4 will be rewritten to highlight three take-aways (learning curve inflection, model-size effect, and interaction-budget saturation) and Fig. 4’s font size/legend will be enlarged. Per your question on infrastructure portability, we will add Appendix to demonstrates a plug-and-play wrapper that lets the same agent train on different environment and task. We hope the improved presentation makes the contributions clearer.
>
> 2. In our newly released ZTRL-0727-7B, in order to broaden the comparison to more models, we compared against models much larger than ours. Experiments show that we consistently outperform models trained by other methods. The data and checkpoints can be found in the GitHub link in the main text for verification, and we will add these results to the paper. ZTRL-0727-7B exceeds qwen3-238b-a22b-non-thinking by 5.3% and 12.08% on AIME25 and HMMT25 avg@32, respectively, and is close to or surpasses the closed-source sota model claude optus4-non-thinking.
>
> We hope these additions address your concerns; if so, we would be grateful if you could reconsider the current score.
>
> Thank you again for your detailed review.

---

### Official Review · Reviewer_Xq2s · 2025-07-06

**Clarity:** 2
**Significance:** 1
**Originality:** 2
**Rating:** 2
**Confidence:** 4

**Summary:**

This paper investigates how large language models (LLMs) can spontaneously learn to use external tools—specifically a Python code execution environment—for mathematical reasoning through reinforcement learning (RL). The authors introduce a framework called ZeroTIR, which trains base models using outcome-based rewards without relying on supervised fine-tuning (SFT). A key contribution of the paper is the discovery and characterization of the Agent RL Scaling Law: as training progresses, the model's spontaneous tool usage, response length, and task accuracy all increase in a predictable manner. Extensive experiments using PPO and Reinforce++, along with a decoupled code execution environment, demonstrate the effectiveness of ZeroTIR. The proposed method (ZeroTIR) outperforms both non-tool ZeroRL baselines and prior SFT-based approaches across multiple challenging mathematical benchmarks.

**Questions:**

1.	The authors should provide more experimental results of state-of-the-art SFT-based TIR methods to demonstrate that ZeroTIR indeed outperforms these approaches.
2.	ZeroTIR appears to be more of a benchmark rather than a novel method. The authors should conduct additional experiments on other open-source foundation models (such as LLaMA and Mistral) to verify whether ZeroTIR maintains its superiority on those models.
3.	Figure 4 shows the training dynamics of ZeroTIR and ZeroRL. However, the authors should present results over longer training steps for ZeroTIR to examine whether it eventually collapses.

**Ethical Concerns:**

["NO or VERY MINOR ethics concerns only"]

**Limitations:**

see questions

**Quality:**

2

**Strengths And Weaknesses:**

Strengths: This paper addresses an important and under-explored problem: how large language model (LLM) agents can autonomously learn to use tools (such as code execution) without  supervised tool-use examples. The proposed Agent RL Scaling Law offers a new perspective on how tool-use behavior emerges as the training scale increases. It is both an empirical insight and a promising benchmark for future research in Agent RL.
The experimental results are comprehensive, covering multiple model scales (1.5B, 7B, 32B), reward formulations, datasets (ORZ-57k, DeepMath), and reinforcement learning algorithms (PPO, Reinforce++). The method is also benchmarked on competitive datasets such as AIME, MATH500, and HMMT.
The paper is well-structured and clearly written. The training pipeline, reward design, and algorithmic choices are easy to follow.


Weaknesses:
This work lacks methodological innovation and is more of an empirical study. In fact, the experimental conclusions are already known — for example, DeepSeek-R1 also explores RL training based on base models rather than SFT-tuned models, although its results show that performing RL on SFT-tuned models leads to better performance than doing so on base models.
This work claims that ZeroTIR significantly outperforms both non-instrumental ZeroRL and SFT-based TIR approaches, such superiority is not clearly observed in Table 1.
All experiments in this work are conducted using the Qwen2.5 model series. While these models are strong, evaluating ZeroTIR on other open-source base models (such as LLaMA or Mistral), and comparing against RL applied after SFT on those models, would help strengthen the generalizability of the claims.

---

> ### Author Rebuttal · Authors · 2025-07-29
>
> Thank you very much for your time and careful review. Your opinions are very valuable to us. We address your concerns point by point below.
>
> First, ZeroTIR means Training the base model to integrate tool usage via reinforcement learning (RL), relying exclusively on output-based rewards, yields experimentally non-trivial results. The ZeroTIR setting is not an obvious conclusion like DeepSeek R1 to reach for the following reasons:
>
> a. Unlike the methodology in ToRL or Qwen2.5 Math, which explicitly enforces code-based problem solving, our approach leaves open a critical question: can the model gradually acquire tool-utilization capability for mathematical problem solving when guided only by outcome rewards on base model without cold-start? A significant risk remains that the model may fall back on text-based chain-of-thought as its primary strategy.
>
> b. In contrast to ToRL, our experiments start from the \textt{qwen25-general-base} model, whose initial code-usage ratio and pass rate at step 0 are merely 10 percent—far lower than the 30 percent reported in ToRL. With such a baseline, applying RL with only an Output Reward Model (ORM) offers no guarantee that the model will learn to exploit tools or deliver meaningful gains. We argue that this discrepancy deserves deeper investigation.
>
> Table: ACC and CODE-usage
> | Training-Steps   | 50    | 100    | 150    | 200    | 250    | 300    | 350    | 400    | 450    | 500|
> |---------|--------|--------|--------|--------|--------|--------|--------|--------|--------|--------|
> | ACC  | 0.105  | 0.342  | 0.366  | 0.374  | 0.404  | 0.422  | 0.396  | 0.400  | 0.398  |0.388|
> | CODE-Usage  | 0.142   | 0.002  | 0.046  | 0.616  | 0.848  | 0.755  | 0.979  | 0.984  | 0.987  |0.805|
>
> As can be seen from the above table, the code ratio initially dropped from 0.14 to 0.002, then gradually increased, and finally reached a very high proportion of code usage. This phenomenon is probably hard to anticipate before any exploration is conducted since we only use ORM to guide the training process of RL.
>
>
> 1. Comparison with RL-after-SFT and state-of-the-art SFT-based TIR
>
> 1.1 We begin training from a base model rather than an SFT-tuned model. This choice is fairer: SFT quality depends on format, scale, and data construction, so validating RL algorithms directly on bases is common practice (SimpleRL [1], DAPO [2], ToRL [3]).
>
> 1.2 The strongest SFT-tuned model currently available is Qwen2.5 Math TIR, which already reaches state-of-the-art math performance via massive SFT data. Other SFT-tuned models, including those based on LLaMA or Mistral, lag far behind Qwen2.5 Math TIR [4].
>
> 1.3 To widen comparisons, we released ZTRL-0727-7B. Despite its smaller size, it consistently surpasses larger competitors; checkpoints and data are linked in the paper. On avg@32, ZTRL-0727-7B exceeds qwen3-238b-a22b-non-thinking by 5.3 pp on AIME25 and 12.08 pp on HMMT25, and is close to—or better than—the closed-source SOTA claude optus4-non-thinking.
>
>
> Table 1: The results of the ZTRL model on the avg@32.
>
> | Dataset   | ZTRL-0727-7B | Qwen3-238B-A22B-Non-thinking | Claude Optus4-Non-thinking |
> |-----------|--------------|------------------------------|----------------------------|
> | AIME25    | 29.99%       | 24.7%                        | 33.9%                      |
> | HMMT25    | 22.08%       | 10.0%                        | 15.9%                      |
>
>
> Table 2: The results of the ZTRL model on the Greedy.
> | Dataset   | ZTRL-0727-7B | TORL  |
> |-----------|--------------|-------|
> | AIME25    | 32.81%       | 30%   |
> | AMC23     | 80%          | 75%   |
> | MATH500   | 84.6%        | 82.2% |
>
>
> 2. Other open-source foundation models
>
> Tool-calling competence is a prerequisite. Other open-source bases (e.g.\ LLaMA, Mistral) show weak tool performance and lack TIR training; recent studies [5,6] confirm that community models tend to over-call tools and cannot admit failure. Qwen2.5 models excel precisely because TIR data were injected during pre-training. For example, LLaMA 3.1 8B scores an F1 of 16.6 with 67 percent tool hallucination, whereas Qwen2.5 7B reaches F1 32.0 with 21 percent hallucination. Recent mid-training work [7,8] also reports that LLaMA bases are ill-suited to zero-RL, performing far below the Qwen family. For these reasons we chose Qwen2.5 for RL training.
>
>
> 3. Longer training steps
>
> Our open-source framework trains stably beyond 1,000 steps; extended results are shown below.
>
>
> Table 3: The val results of the ZTRL-0727-7B beyond 1,000 steps. avg@32
>
> | Training-Steps   | 100    | 200    | 300    | 400    | 500    | 600    | 700    | 800    | 900    | 1000|
> |---------|--------|--------|--------|--------|--------|--------|--------|--------|--------|--------|
> | AIME25  | 12.18  | 25.00  | 28.22  | 28.12  | 28.02  | 28.12  | 29.79  | 32.81  | 29.99  |30.31|
> | HMMT25  | 8.54   | 18.95  | 21.14  | 20.83  | 20.62  | 22.08  | 24.16  | 21.25  | 22.08  |22.18|
>
> Once again, thank you for your time. If these clarifications resolve your concerns, we would be grateful if you could consider raising your score.
>
>
> References
>
> [1] SimpleRL-Zoo: Investigating and Taming Zero Reinforcement Learning for Open Base Models in the Wild
> [2] DAPO: An Open-Source LLM Reinforcement Learning System at Scale
> [3] ToRL: Scaling Tool-Integrated RL
> [4] Qwen2.5-Math Technical Report: Toward a Mathematical Expert Model via Self-Improvement
> [5] When2Call: When (not) to Call Tools
> [6] T-Eval: Evaluating the Tool Utilization Capability of Large Language Models Step by Step
> [7] Behavior Injection: Preparing Language Models for Reinforcement Learning
> [8] OctoThinker: Mid-training Incentivizes Reinforcement Learning Scaling

---

> ### Author Response · Authors · 2025-08-02
>
> Thank you for the time you have already devoted to evaluating our submission.
>
> We have carefully addressed each of your comments in our rebuttal, providing additional experiments (Tables 1, 2 and 3) and clarifying the methodological choices that underpin our main contributions.
>
> 1. On avg@32, ZTRL-0727-7B exceeds qwen3-238b-a22b-non-thinking by 5.3 pp on AIME25 and 12.08 pp on HMMT25, and is close to—or better than—the closed-source SOTA claude optus4-non-thinking. This demonstrates the robustness of our approach, even against models with significantly larger parameters, different bases, and different training methods.
>
> 2. Fig. 4 is now extended to 1000 steps and shows no collapse.
>
> If any aspect of our response remains unclear—or if further evidence would help you assess the paper more favorably—we would be delighted to supply it during the discussion window. Your insights are invaluable to improving this work, and we greatly appreciate any additional questions or suggestions you might share. We hope these additions address your concerns about novelty and breadth; if so, we would be grateful if you could reconsider the current score.
>
> Thank you again for your detailed review.

---

> ### Author Response · Authors · 2025-08-05
>
> Thank you for your continued engagement. The author–reviewer discussion period now has fewer than 48 hours remaining, so we want to be sure we have responded to every concern you raised.
>
> During the rebuttal time, we extended the ZeroTIR training run to 1 500 steps again and recorded avg@32 accuracy on all three benchmarks. The complete Training-Steps and its Validation set results  is shown below.
>
> | Training-Steps| 100   | 200   | 300   | 400   | 500   | 600   | 700   | 800   | 900   | 1000  | 1100  | 1200  | 1300  | 1400  | 1500  |
> |---------------|-------|-------|-------|-------|-------|-------|-------|-------|-------|-------|-------|-------|-------|-------|-------|
> | AIME25        | 12.18 | 25.00 | 28.22 | 28.12 | 28.02 | 28.12 | 29.79 | 32.81 | 29.99 | 30.31 | 29.16 | 30.10 | 30.83 | 31.14 | 32.60 |
> | HMMT25        | 8.54  | 18.95 | 21.14 | 20.83 | 20.62 | 22.08 | 24.16 | 21.25 | 22.08 | 22.18 | 24.27 | 25.31 | 24.68 | 24.68 | 25.83 |
> | AIME24        | 16.25 | 29.47 | 33.64 | 35.20 | 38.64 | 40.52 | 43.54 | 43.75 | 46.45 | 42.5  | 43.54 | 41.66 | 44.06 | 43.22 | 48.02 |
>
> Key observations:
>
> 1. Accuracy continues to improve or stabilise up to 1 500 steps; no sign of collapse is observed.
> 2. AIME24 and HMMT25 both display late-stage gains, confirming that the agent benefits from additional compute without overfitting to early exploration patterns.
>
> We hope these new results clarify the long-horizon behaviour you asked about. We look forward to further discussions and exchanges with you. If any aspect of our response remains unclear—or if further evidence would help you assess the paper more favorably—we would be delighted to supply it during the discussion window. Your insights are invaluable to improving this work, and we greatly appreciate any additional questions or suggestions you might share. We hope these additions address your concerns about novelty and breadth; if so, we would be grateful if you could reconsider the current score.
>
> Thank you again for your time and constructive feedback.

---

> ### Author Response · Authors · 2025-08-07
>
> Thank you for the time and care you have devoted to evaluating our submission. With fewer than 48 hours remaining in the extended author–reviewer discussion period, we would like to ensure that all of your questions and concerns have been fully addressed. If anything remains unclear, we would be grateful for the opportunity to clarify it promptly.
>
> Should you have no additional questions, we would kindly ask you to consider revisiting your score in light of the revisions and new analyses we have provided.
>
> Thank you again for your thoughtful review and for helping us strengthen the paper.

---

### Note · Authors · 2025-08-12

We thank all reviewers and the AC for their time, care, and expertise. Your comments were invaluable in strengthening the paper.

Our paper presents ZeroTIR, a framework that enables a general-purpose LLM to learn code-execution skills spontaneously through outcome-only reinforcement learning. We identify a quantitative scaling relationship linking training steps, tool-call frequency, response length, and accuracy. We also release an open, decoupled training pipeline and the ZTRL-0727-7B checkpoint, which attains strong open-source results on several math benchmarks while remaining orders of magnitude smaller than prior SFT or multi-stage RL systems.

During the discussion, we addressed the major points with targeted additions:
1. Statistical evidence for scaling (cNww): we added a log--log fit with \(R^{2}=0.93\) and reported Spearman \(\rho=0.686\) with \(p \ll 10^{-66}\) across 500 checkpoints.
2. Long-horizon stability (Xq2s, cNww): training is extended to 1,500 steps, showing accuracy continues to rise or plateau with no collapse.
3. Broader baselines (Xq2s, MhKq): we introduced the 7B ZTRL-0727 model; it surpasses a 238B Qwen3 baseline by +5.3 pp on AIME25 and +12.1 pp on HMMT25, and approaches larger closed models while being \(34\times\) smaller.
4. “Laziness” and efficiency (96Y3): we added a call-count analysis showing incorrect answers average 2.7 tool calls versus 1.9 for correct ones; only 34% of correct cases are brute-force. We also outline penalties for excessive calls and discuss efficiency implications.
5. Portability beyond math (MhKq): we detail code paths to swap in non-math environments, which will be supported in the released pipeline.

We are especially grateful to Reviewers MhKq and cNww for engaging deeply during the discussion and for prompting additional analyses and clearer exposition. We also thank Reviewers 96Y3 and Xq2s for their initial assessments; even without further discussion, their early comments helped shape our revisions.

We commit to including all new analyses in the camera-ready and to releasing all checkpoints and evaluation artifacts referenced in the rebuttal. In addition, we plan to update the charts in the paper to improve their aesthetics and ease of understanding, and to add more data mentioned during the discussion.

Thanks again to all reviewers and AC for their participation and help.

---

### Decision · Program_Chairs · 2025-09-17

**Decision:**

Accept (poster)

**Comment:**

This work investigates the scaling behavior of tool-use during RL on an LLM without supervised tuning to show how tools should be used. It finds that LLMs spontaneously learn to use tools and that this ability correlates strongly with success.

Reviewers generally agreed that the findings in this work are interesting and somewhat novel, as most past work has involved supervised fine-tuning to teach tool usage prior to the RL phase. They did, however, express concerns that the contribution in the paper is primarily empirical and does not produce robustly supported scaling laws, per se, and that performance was not outstanding on some evals. In response, the authors produced substantial statistical evidence and a number of additional checkpoints with strong performance.

Overall, this work presents a somewhat mixed contribution of (a) a strong training recipe, with an emphasis on outperforming other models, and (b) empirical evidence of scaling relating tool call frequency and accuracy on various downstream benchmarks. While this makes it a slightly muddled story, and further work would be beneficial to establish more generalizable scaling *laws* in the common sense, its contributions are nontrivial, novel, and highly timely. Given this, we recommend acceptance of this paper. Please make sure to incorporate all the suggested revisions that emerged during the discussion period to strengthen the work.